# Long-Term Monoculture Negatively Regulates Fungal Community Composition and Abundance of Tea Orchards

Yasir Arafat [1,2,3,†] , Muhammad Tayyab [4,5,6,†], Muhammad Umar Khan [1,2,3], Ting Chen [1,2,3], Hira Amjad [1,2,3], Saadia Awais [3,5,6,7], Xiangmin Lin [1,2,3,*] , Wenxiong Lin [1,2,3,5,6,*] and Sheng Lin [1,2,3,5,6,*]

1   College of Life Science, Fujian Agriculture and Forestry University, Fuzhou 35002, China
2   Key Laboratory of Fujian Province for Agroecological Process and Safety Monitoring, Fujian Agriculture and Forestry University, Fuzhou 35002, China
3   Institutes of Agroecology, Fujian Agriculture and Forestry University, Fuzhou 35002, China
4   Key Laboratory of Sugarcane Biology and Genetic Breeding, Ministry of Agriculture, Fujian Agriculture and Forestry University, Fuzhou 350002, China
5   College of Crop Science, Fujian Agriculture and Forestry University, Fuzhou 350002, China
6   Key Laboratory of Genetics, Breeding and Multiple Utlization of Crops, Ministry of Education, Fujian Agriculture and Forestry University, Fuzhou 350002, China
7   Stress Physiology and Molecular Biology Lab, Government College Women University, Faisalabad 38000, Pakistan
*   Correspondence: xiangmin@fafu.edu.cn (X.L.); lwx@fafu.edu.cn (W.L.); linsh@fafu.edu.cn (S.L.)
†   Authors with equal contribution.

**Abstract:** Continuous cropping frequently leads to soil acidification and major soil-borne diseases in tea plants, resulting in low tea yield. We have limited knowledge about the effects of continuous tea monoculture on soil properties and the fungal community. Here, we selected three replanted tea fields with 2, 15, and 30 years of monoculture history to assess the influence of continuous cropping on fungal communities and soil physiochemical attributes. The results showed that continuous tea monoculture significantly reduced soil pH and tea yield. Alpha diversity analysis showed that species richness declined significantly as the tea planting years increased and the results based on diversity indicated inconsistency. Principal coordinate analysis (PCoA) revealed that monoculture duration had the highest loading in structuring fungal communities. The relative abundance of *Ascomycota*, *Glomeromycota*, and *Chytridiomycota* decreased and *Zygomycota* and *Basidiomycota* increased with increasing cropping time. Continuous tea cropping not only decreased some beneficial fungal species such as *Mortierella alpina* and *Mortierella elongatula*, but also promoted potentially pathogenic fungal species such as *Fusarium oxysporum*, *Fusarium solani*, and *Microidium phyllanthi* over time. Overall, continuous tea cropping decreased soil pH and potentially beneficial microbes and increased soil pathogenic microbes, which could be the reason for reducing tea yield. Thus, developing sustainable tea farming to improve soil pH, microbial activity, and enhanced beneficial soil microbes under a continuous cropping system is vital for tea production.

**Keywords:** continuous cropping obstacle; soil acidification; soil-borne diseases; pathogenic microbes; beneficial microbes

## 1. Introduction

Tea (*Camellia sinensis* L.) is an important traditional and cash crop extensively grown in southern China, Asia, and Africa. It was introduced to China over 1500 years ago [1,2]. Tea has become

increasingly famous owing to its pleasantly mild taste and well-known health benefits such as anticancer, anti-oxidation and anti-inflammatory properties [3]. The limited tea-planting area, as well as an increasing market requirement, frequently lead to the implementation of consecutive monoculture tea-growing regimes in China. However, long-term monoculture frequently leads to soil acidification and major soil-borne diseases in tea plants, thereby limiting tea yield [2]. Thus, the reduction in tea production not only stops the sustainable development of the Chinese tea industry, but also causes rapid economic losses. Consequently, these leading obstacles attributed to the long-term monoculture of tea have attracted the attention of many soil ecologists and scientists [1,4,5].

Consecutive monoculture problems, identified as replanting diseases, have generally occurred in both perennials and annual crops such as coffee, sugarcane, apple, potato and soybean [6–10]. Consecutive monoculture problems are commonly attributed to the stimulation of soil-borne pathogens, the reduction of soil physiochemical features, and the accretion of autotoxic substances [11–13]. Research has revealed that shifts in soil microbiome, in terms of accumulation in fungal pathogens, are often responsible for continuous monoculture problems [14,15]. For example, Song et al. [16] study revealed that *Coptis chinensis* monoculture altered the soil overall fungal communities by magnifying soil-borne pathogens such as *Fusarium*, which, in turn, promoted root rot disease as well as limited the crop yield. However, the general shift in fungal composition, especially in potentially beneficial and pathogenic fungi, has not been studied during continuous tea cropping. Therefore, we assume that the consecutive monoculture of tea may have a direct influence on soil physicochemical characteristics and fungal communities, in turn adversely affecting tea yield. In this study, we considered the fungal communities in cultivated tea fields with 2, 15, and 30 years of monoculture history. The objectives of our study were to (a) assess the soil physiochemical attributes and soil fungal communities, including the fungal diversity and the compositional and structural changes associated with the different continuous cropping histories of the tea fields, (b) explore the underlying relationships among the dominant soil fungal taxa (phylum, genus, and species) and soil physiochemical attributes of tea monocultures, and (c) discuss the implications for the relationship between soil physical parameters and the resultant fungal community in terms of potential to influence tea yields.

## 2. Materials and Methods

### 2.1. Site Description and Soil Sampling

The trial site was Fujian Agriculture and Forestry University tea fields, Fuzhou City, Fujian Province, China (latitude: 26°05′9.60″ N; longitude: 119°14′3.60″ E) (27°43′ N, 118°72′ E). The area had been fertilized annually with the following: 30 kg/Mu of urea (65% N), 6 kg/Mu of superphosphate (13.04% $P_2O_5$), and 10 kg/Mu of potash (1.74% $K_2O$). The annual temperature and average yearly precipitation were 20–25 °C and 900–1362 mm, respectively. The soil of this area was clay loam [17]. Rhizosphere soil samples were collected from different age monoculture tea fields (i.e., 2TY, tea grown for 2 years; 15TY, tea grown for 15 years; and 30TY, tea grown continuously for 30 years) in April 2016. Furthermore, the bulk soil (labeled "CK") was obtained from the nearby uncultivated field. There were 2 or 3 lines of tea trees in each field. The space between the lines was 150 cm and the space between each tree was 30 cm. Tea roots were carefully uprooted from the soil with a forked spade and slightly shaken to remove loosely attached rhizosphere soil [4]. Twenty random plants per field were selected and five plants were pooled together as one replication. Thus, each field was sampled with four replications. The rhizosphere soil tightly attached to the roots was brushed off and collected in plastic bags. The soil samples were instantly stored into a sterile icebox and brought to the laboratory. Using 2 mm mesh, all soil samples were sieved and divided into two subsamples; one portion of every sample was air-dried for analyzing physiochemical soil characteristics and the remainder was kept at −80 °C for DNA extraction.

### 2.2. Yield Determination

The yield of fresh and dry leaves was calculated in Kg·ha$^{-1}$ [17].

### 2.3. Measurement of Soil Physiochemical Properties

Soil suspension with water (1:2.5 WV$^{-1}$) was prepared in order to estimate soil pH using a pH meter (PHS-3C, INESA Scientific Instrument Co., Ltd., Shanghai, China). Molybdenum Blue protocol was followed in order to measure available phosphorus (AP) by using hydrochloric acid and ammonium fluoride [2,18]. For the estimation of available nitrogen (AN), the alkaline hydrolyzable method was carried out, while available potassium (AK) was extracted by ammonium acetate and measured by flame photometry [19]. The potassium dichromate internal heating method was used for the measurement of soil organic matter (SOM) [20].

### 2.4. DNA Extraction and PCR Amplification

Using the Fast DNA$^{TM}$ Spin kit (MP Biomedical, Santa Ana, CA, USA), total genomic DNA was extracted from soil samples and then purified with a DNA purification kit (Tiangen Biotech Co., Ltd., Beijing, China) according to the manufacturer's instructions. The quality and quantity of DNA were calculated with NanoDrop (Thermo Fisher Scientific, Middletown, VA, USA) and then kept at −20 °C for sequencing. For the amplification of the fungal ITS1 region, the primers ITS5-1737F and ITS2-110 2043R were used [21]. PCR reactions were directed in 30 μL mixtures with each primer (0.2 μM), DNA templates (10 ng) and Phusion$^®$ High-Fidelity PCR Master Mix (15 μL) (New England BioLabs, Ipswich, MA, USA). The conditions set for PCRs were as follows: 98 °C for one minute, followed by 30 cycles of 98 °C for 10 s, 50 °C for 30 s, and 72 °C for 60 s with a final extension at 72 °C for five minutes. The PCR products were purified using QIAquik Gel Extraction Kit (QIAGEN, Düsseldorf, Germany) and the sequencing libraries were generated using TruSeq$^®$ DNA PCR-Free Sample preparation kit (Illumina, San Diego, CA, USA) and their quantities measured on a Qubit @ 2.0 fluorometer (Thermo Fisher Scientific, Waltham, MA, USA) and the Agilent Bioanalyzer 2100 system (Santa Clara, CA 95051, USA). Lastly, the DNA libraries were sequenced on an Illumina HISeq2500 platform by Novogen (Beijing, China).

### 2.5. Statistical and Bioinformatics Analysis

The paired-end reads recovered from the original DNA fragment were combined using FLASH (Baltimore, MD, USA) [22] and based on the unique barcode assigned to each sample. The sequences were assigned to the same operational taxonomy unit (OTU) based on 97% similarity. For each OTU, the representative sequences were selected and a ribosomal database project (RDP) classifier [23] was employed for annotating the taxonomic information for each representative sequence. Using quantitative insight microecology (QIIME), Simpson, Shannon, Chao1, and ACE indices were examined for measuring alpha diversity and species richness [24–26] and were presented using R software (version 2.15.3, R Foundation for Statistical Computing, Vienna, Austria). The rarefaction curves were constructed based on the observed species richness, and the unique and common OTUs among the soil samples were displayed by Venn diagram. In order to study the changes in species complexity among samples, unweighted UniFrac principal coordinate analysis (PCoA) and unweighted UniFrac unweighted pair group method with arithmetic mean analysis (UPGMA) were carried out. The R Package was used for the redundancy analysis (RDA). The least significant difference (LSD) test was used to evaluate the significant differences of soil physiochemical properties by DPS software (version 7.05, Elite Law Solicitors, Amersham, London, UK). Additionally, Pearson correlation coefficients among fungal abundance and soil physiochemical properties were carried out.

## 3. Results

### 3.1. Yield of Tea Leaves in Different Age Monoculture Tea Fields

The tea production from different age monoculture of tea fields is given in Table 1. As compared with the newly planted field (T2Y), the fresh leaves yield of T15Y and T30Y was significantly reduced by 2.88% and 4.50%, respectively. Further, the dry leaves yield of T15Y and T30Y was significantly reduced by 6.28% and 15.34%, respectively.

**Table 1.** Yield of Tea Leaves in Different Age Monoculture Tea Fields.

| | (Yield per unit) | |
| --- | --- | --- |
| Treatment | (Fresh weight) | (Dry weight) |
| | ——kg·ha$^{-1}$—— | |
| T2Y | 1436.37 ± 5.54 [a] | 408.33 ± 2.03 [a] |
| T15Y | 1395.00 ± 1.73 [b] | 382.67 ± 1.75 [b] |
| T30Y | 1371.67 ± 5.55 [c] | 345.67 ± 2.03 [c] |

Note: T2Y, T15Y, and T30Y represent fields consecutively planted for 2, 15, and 30 years, respectively. Values are means ± standard deviations and different letters within the same column denoted a significant difference at $p > 0.05$.

### 3.2. Soil Physicochemical Characteristics from Different Age Monoculture Tea Fields

The physiochemical characteristics of soil from control and different age monoculture of tea fields are given in Table 2. Overall, the soil pH significantly declined with the continuous cropping of tea. Compared with the newly planted field (T2Y), the continuously cropped fields (T15Y and T30Y) significantly reduced the AN. The T15Y plantation soil significantly decreased the SOM and AP compared to the more recent and older planted fields (T2Y and T30Y). However, there was a non-significant difference for AK between the control and different age monoculture tea fields.

**Table 2.** Physiochemical characteristics of soil under continuous tea cropping.

| Samples | pH | SOM (g kg$^{-1}$) | AN (mg kg$^{-1}$) | AK (mg kg$^{-1}$) | AP (mg kg$^{-1}$) |
| --- | --- | --- | --- | --- | --- |
| CK | 5.07 ± 0.06 [a] | 12.20 ± 0.37 [b] | 13.09 ± 0.10 [c] | 132.79 ± 2.69 [a] | 85.72 ± 0.52 [c] |
| T2Y | 4.23 ± 0.06 [b] | 13.45 ± 0.32 [a] | 14.69 ± 0.10 [a] | 130.23 ± 0.87 [a] | 91.19 ± 0.42 [a] |
| T15Y | 3.58 ± 0.28 [c] | 12.39 ± 0.26 [b] | 13.74 ± 0.10 [b] | 128.48 ± 0.53 [a] | 88.04 ± 0.80 [b] |
| T30Y | 3.32 ± 0.08 [c] | 13.74 ± 0.05 [a] | 13.95 ± 0.09 [b] | 133.24 ± 0.59 [a] | 92.40 ± 0.71 [a] |

Note: CK, T2Y, T15Y, and T30Y represent fields consecutively planted for 0, 2, 15, and 30 years, respectively. SOM, soil organic matter; AN, available nitrogen; AK, available potassium; AP, available phosphorus. Values are means ± standard deviations and different letters within the same column denoted a significant difference at $p > 0.05$.

### 3.3. Fungal Alpha Diversity and Species Richness

A total of 1,165,375 (average: 72,836) reads were obtained from all the samples (Figure S1, Supplementary Materials). According to rarefaction analysis curve, the OTU numbers for ITS were plateaued at 97% similarity after 45,000 sequences (Figure 1A). This confirmed that the depth of sequencing was appropriate in terms of capturing soil fungal richness and diversity from control and different age monoculture tea fields. A total of 1790, 1495, 978, and 1228 OTUs were found in soil samples obtained, respectively, from CK, T2Y, T15Y, and T30Y (Figure 1B). The fungal community richness (Chao1, observed species, and ACE indices) decreased significantly in response to continuous cropping of tea (Figure 2). In comparison with CK and T15Y, alpha diversity indices, such as Simpson and Shannon, increased significantly in the fresh and older tea plantation (T2Y and T30Y) (Figure 2D,E). These results indicate that species richness declines significantly as the tea planting years increase and the results based on diversity indicate inconsistency.

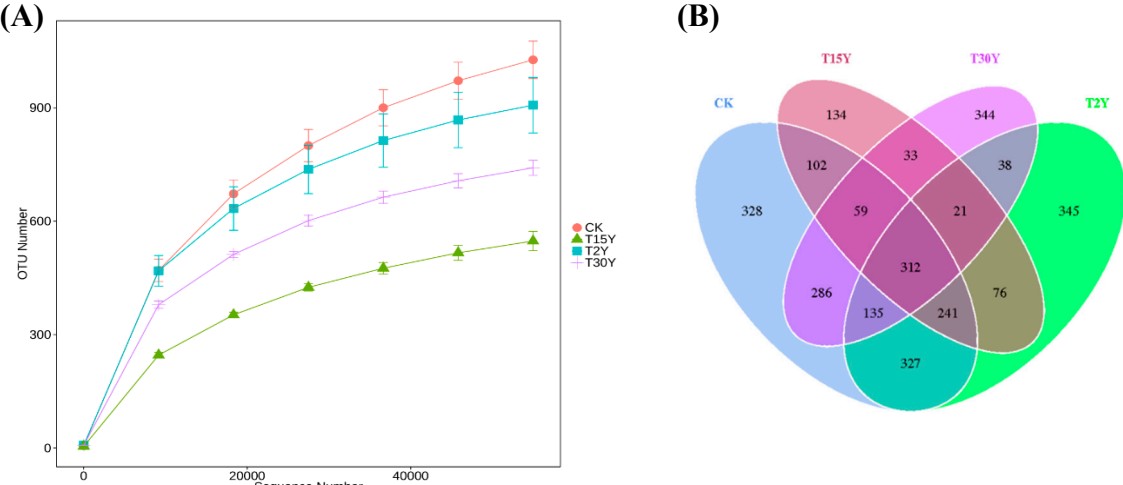

**Figure 1.** (**A**) Rarefaction curves with operational taxonomy unit (OTU) threshold of 97% sequence similarity and (**B**) Venn diagram for control and different age monoculture tea fields.

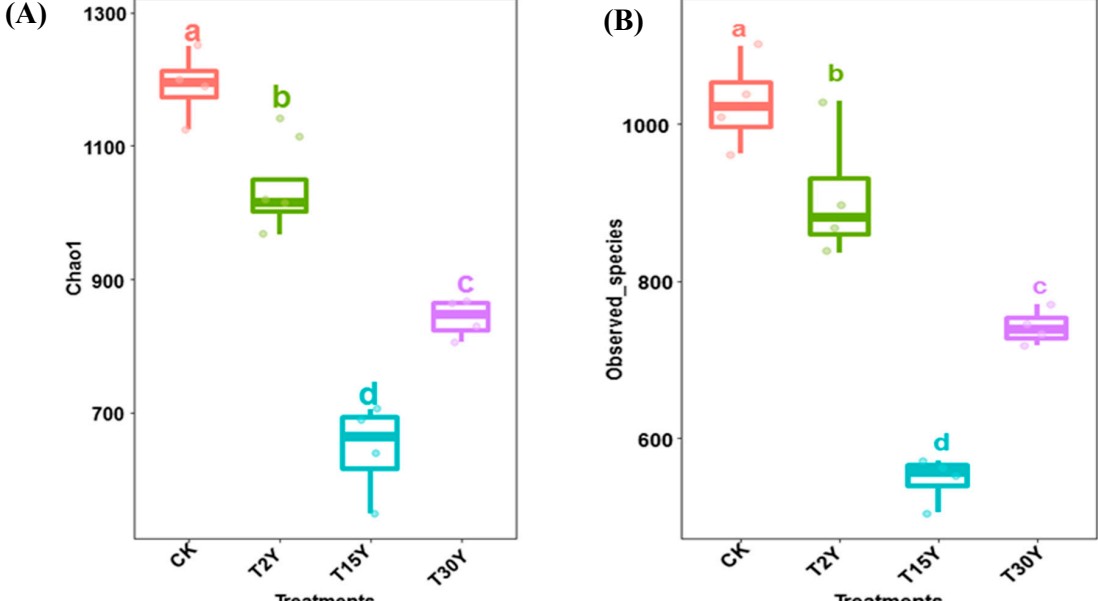

**Figure 2.** *Cont.*

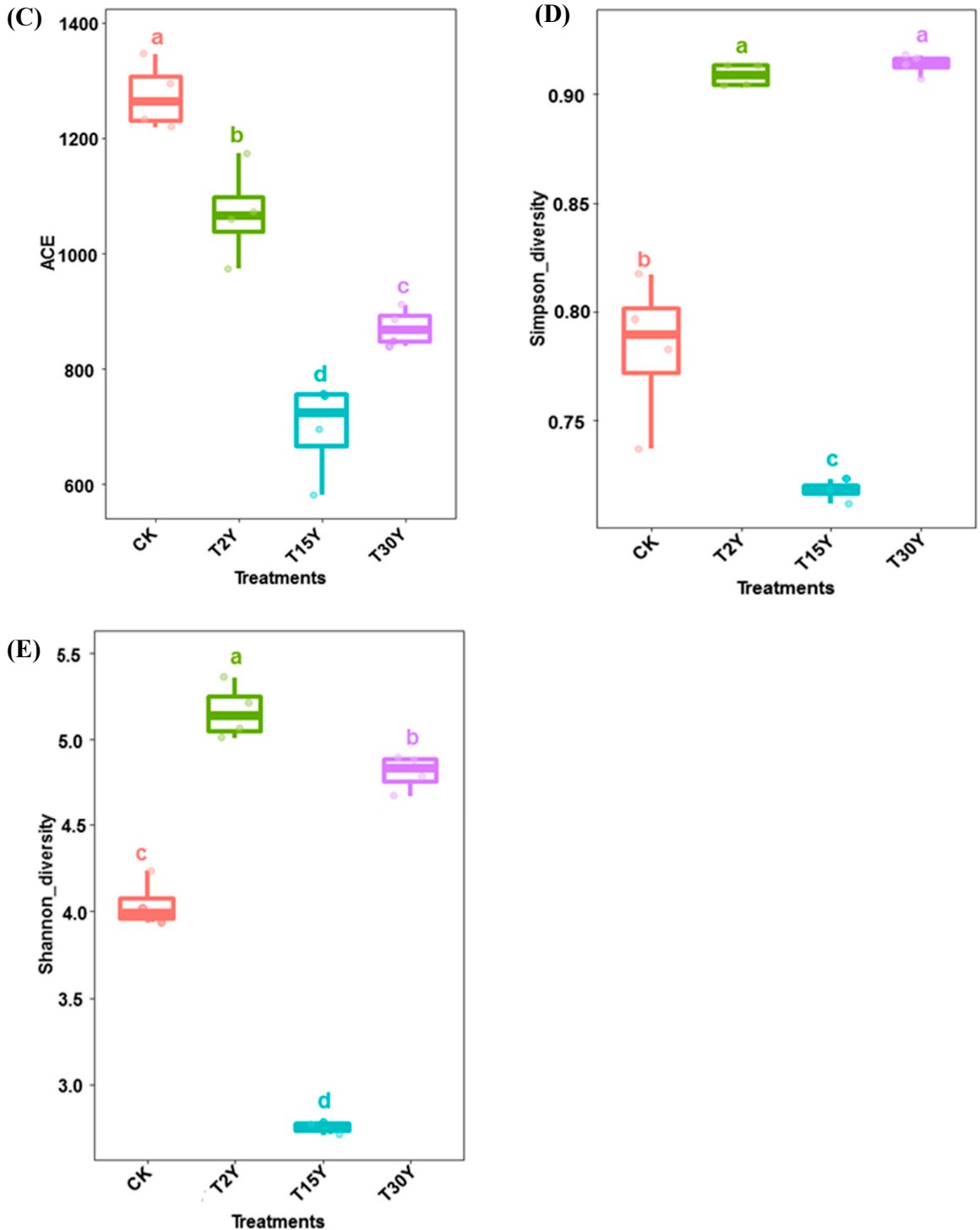

**Figure 2.** Box plots that represent alpha diversity indices such as fungal community richness (Chao1, observed species, ACE) and diversity (Shannon, Simpson) in the control and different age monoculture tea fields. (**A**) Chao; (**B**) observed species; (**C**) Ace; (**D**) Simpson; (**E**) Shannon. Error bars with different lowercase letters indicate significant differences among treatments based on the LSD test (*** $p < 0.05$). CK, T2Y, T15Y, and T30Y represent fields consecutively planted for 0, 2, 15, and 30 years, respectively.

### *3.4. Fungal B-Diversity*

Unweighted UniFrac (UUF) principal coordinate analysis (PCoA) showed distinct fungal group patterns in the control and different age monoculture tea fields, with the first and second axes depicting 46.89% of the complete change in fungal data (Figure 3A). The unweighted pair group method with

arithmetic mean analysis (UPGMA) and non-metric multidimensional scaling (NMDS) analysis further confirmed that samples derived from control and different age monoculture tea fields were separated (Figure 3B; Figure S2, Supplementary Materials). These results indicate that the soil fungal communities are strongly influenced by long-term consecutive monoculture.

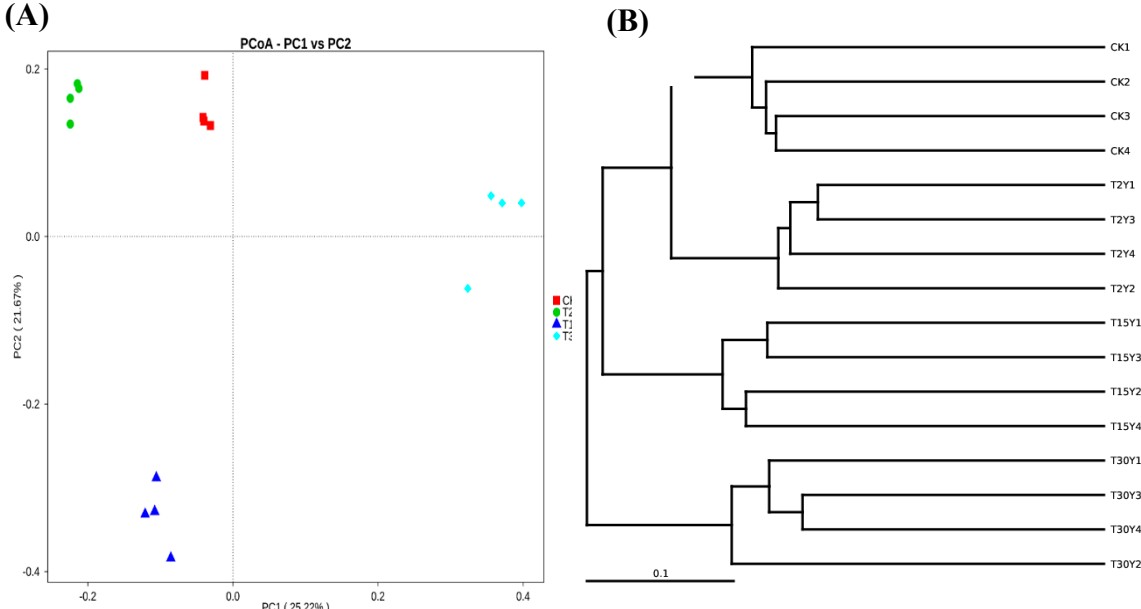

**Figure 3.** (**A**) Unweighted UniFrac (UUF) principal coordinate analysis (PCoA) and (**B**) unweighted pair group method with arithmetic mean analysis (UPGMA) of fungal communities in control and different age monoculture tea fields. CK, T2Y, T15Y, and T30Y represent fields consecutively planted for 0, 2, 15, and 30 years, respectively.

### 3.5. Relative Abundance of Fungal Communities From Different Age Monoculture Tea Fields

There were different changes in fungal phyla among control samples as well as different age monoculture tea fields (CK, T2Y, T15Y, and T30Y). *Ascomycota*, *Zygomycota*, *Basidiomycota*, *Glomeromycota*, and *Chytridiomycota* were the dominant phyla in different age monoculture tea fields, including CK (Figure 4). Continuous cropping of old tea field (T15Y) significantly reduced the abundance of *Ascomycota*, *Glomeromycota*, and *Chytridiomycota*, respectively. The relative abundance of *Zygomycota* was significantly higher in old planted fields (T15Y and T30Y) rather than control and freshly grown fields (CK and T2Y). In comparison with CK and T15Y, the abundance of *Basidiomycota* in T2Y and T30Y was significantly higher (Figure S2, Supplementary Materials). The table was created to fully and directly demonstrate the variability, similarities, and relative abundance of fungal compositions in samples obtained from four fields. The ten most predominant fungal genera and species were sketched as a heat map diagram in four samples (Figure 5). Among them, highly abundant fungal genera such as *Mortierella*, *Microidium*, *Fusarium*, *Dactylonectria*, *Gibberella*, *Ilyonectria*, *Lycogalopsis*, *Melanconiella*, and *Phoma* were found in all samples (Table S2, Supplementary Materials).

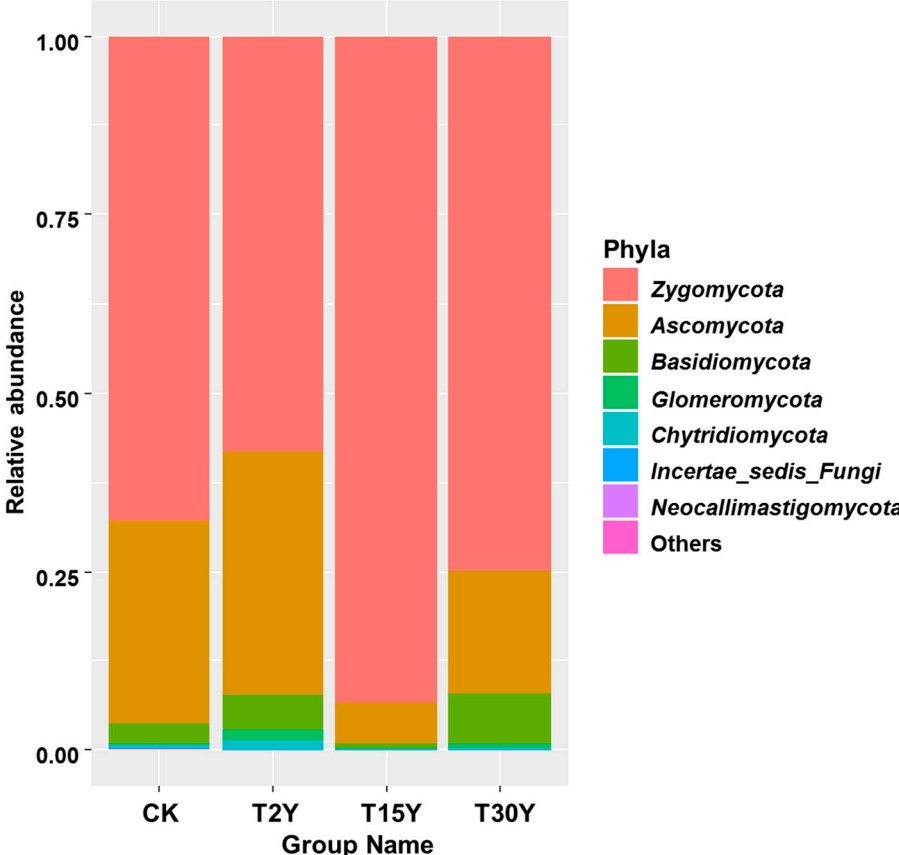

**Figure 4.** Relative abundance of top fungal phyla in control and different age monoculture tea fields. "Others" shows the sum of the relative abundances of all phyla except the eight listed.

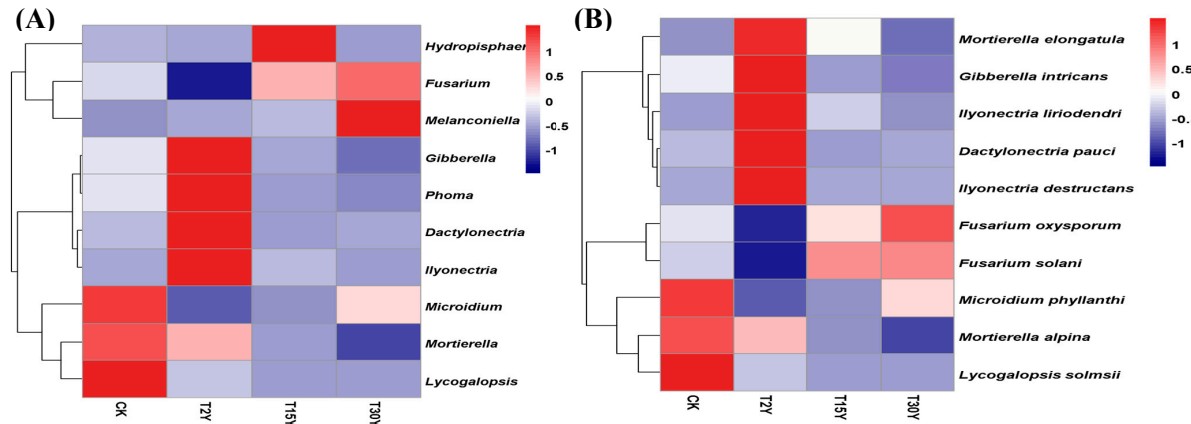

**Figure 5.** Composition and distribution of 10 dominant (**A**) genera and (**B**) species in soil obtained from control and different age monoculture tea fields. The color code displays relative abundance, ranging from red (high abundance) to blue (low abundance). CK, T2Y, T15Y, and T30Y represent fields consecutively planted for 0, 2, 15, and 30 years, respectively.

Compared to the newly planted field (T2Y), continuously planted fields (T15Y and T30Y) had significantly reduced abundance of specific genera (e.g., *Mortierella*, *Dactylonectria*, *Gibberella*, *Ilyonectria*, *Lycogalopsis*, and *Phoma*), whereas *Microidium* and *Fusarium* increased significantly (Figure 5A; Figure S3, Supplementary Materials). Similarly, species belonging to these genera showed similar trends in the newly planted (T2Y) and continuously growing (T15Y and T30Y) fields. For example, compared to the newly planted field (T2Y), the fields planted continuously (T15Y and T30Y) had significantly reduced abundance of some species (e.g., *Mortierella alpina*, *Mortierella elongatula*, *Dactylonectria pauciseptata*,

*Gibberella intricans*, *Ilyonectria liriodendri*, *Ilyonectria destructans*, *Lycogalopsis solmsii*), whereas *Fusarium oxysporum* and *Fusarium solani* increased significantly (Figure 5B; Figure S4, Supplementary Materials). Pearson's correlation coefficients among microbial taxa and time of tea monoculture (T2Y, T15Y, and T30Y) further confirmed these results (Table S2, Supplementary Materials).

### 3.6. Effects of Soil Physiochemical Properties on Fungal Communities

Redundancy analysis (RDA) was performed in order to quantify the effect of soil factors on fungal community composition. The results showed different patterns in fungal communities by soil physiochemical properties, with the first and second axes explaining 65.49.% and 9.44% of the total shift in fungal data, respectively. In addition to these, SOM, AP, and pH were dominant, whereas AN and AK were minor factors in terms of shifting fungal community composition during continuous tea cropping (Figure 6). Pearson's correlation was further used to evaluate relationships between abundant fungal taxa at low taxonomic levels (e.g., genus and species) and soil physicochemical properties (Table 3). Soil pH was significantly and positively correlated with *Ascomycota*, *Neocallimastigomicota*, *Mortierella* (*Mortierella alpina*), and *Lycogalopsis* (*Lycogalopsis solmsii*), and a negative correlation was observed for *Zygomycota* and *Incertae sedis Fungi*, *Melanconiella*, and *Fusarium solani*. However, *Basidiomycota*, *Glomeromycota*, and *Melanconiella* correlated significantly and positively with SOM and AP, whereas *Mortierella* and *Mortierella alpina* were significantly and negatively correlated with AP. *Glomeromycota*, *Chytridiomycota*, *Dactylonectria* (*Dactylonectria pauciseptata*), *Gibberella* (*Gibberella intricans*), *Ilyonectria* (*Ilyonectria liriodendri*, *Ilyonectria destructans*), *Phoma*, and *Mortierella elongatula* were significantly and positively correlated with AN (Table 3).

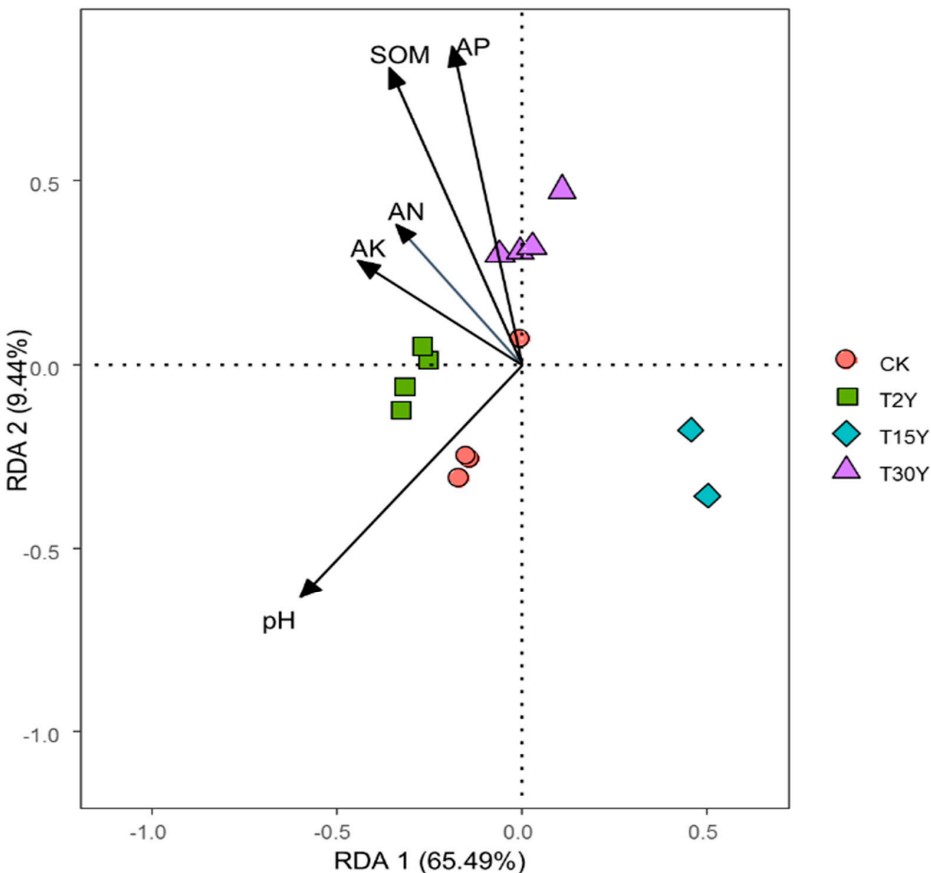

**Figure 6.** Redundancy analysis (RDA) based on the relative abundances of fungal phyla and soil attributes ((pH, SOM, AN, AP, and AK) for individual samples derived from the control and different age monoculture tea fields. CK, T2Y, T15Y, and T30Y represent tea fields with 0, 2, 15, and 30 years of continuous cropping history, respectively.

**Table 3.** Pearson's correlation between soil properties and abundant taxa of fungi (phyla, genera, and species) in control and different age monoculture tea fields.

| Fungal abundant phyla | pH | SOM | AN | AK | AP |
|---|---|---|---|---|---|
| *Zygomycota* | **−0.50 \*** | −0.32 | −0.34 | −0.36 | −0.17 |
| *Ascomycota* | **0.61 \*** | 0.15 | 0.24 | 0.33 | −0.00 |
| *Basidiomycota* | −0.26 | **0.79 \*\*** | 0.39 | 0.45 | **0.75 \*\*** |
| *Glomeromycota* | 0.08 | **0.60 \*** | **0.7 \*\*** | −0.03 | **0.54 \*** |
| *Chytridiomycota* | 0.37 | 0.32 | **0.60 \*** | −0.11 | 0.24 |
| *Incertae_sedis_Fungi* | **−0.49 \*** | 0.47 | 0.04 | 0.33 | 0.49 |
| *Neocallimastigomycota* | **0.61 \*** | −0.10 | −0.103 | −0.00 | −0.26 |
| Fungal abundant genera | | | | | |
| *Mortierella* | **0.94 \*\*** | −0.41 | −0.23 | 0.05 | **−0.58 \*** |
| *Microidium* | 0.36 | −0.11 | −0.49 | 0.26 | −0.22 |
| *Fusarium* | −0.45 | −0.08 | −0.33 | 0.16 | −0.06 |
| *Dactylonectria* | 0.21 | 0.35 | **0.77 \*\*** | −0.16 | 0.34 |
| *Gibberella* | 0.37 | 0.17 | **0.62 \*\*** | −0.15 | 0.13 |
| *Ilyonectria* | 0.14 | 0.35 | **0.80 \*\*** | −0.27 | 0.36 |
| *Lycogalopsis* | **0.63 \*\*** | −0.26 | −0.43 | 0.09 | −0.45 |
| *Melanconiella* | **−0.64 \*\*** | **0.53 \*** | 0.12 | 0.31 | **0.51 \*** |
| *Phoma* | 0.35 | 0.21 | **0.64 \*\*** | −0.12 | 0.18 |
| *Hydropisphaera* | −0.34 | −0.45 | −0.14 | **−0.60 \*** | −0.31 |
| Fungal abundant species | | | | | |
| *Mortierella alpina* | **0.94 \*\*** | −0.40 | −0.24 | 0.09 | **−0.58 \*** |
| *Microidium phyllanthi* | 0.36 | −0.11 | −0.49 | 0.26 | −0.22 |
| *Dactylonectria pauciseptata* | 0.21 | 0.35 | **0.77 \*\*** | −0.16 | 0.34 |
| *Fusarium oxysporum* | −0.39 | −0.00 | −0.25 | 0.22 | −0.00 |
| *Gibberella intricans* | 0.38 | 0.18 | **0.62 \*\*** | −0.13 | 0.14 |
| *Fusarium solani* | **−0.52 \*** | −0.12 | −0.36 | 0.10 | −0.08 |
| *Mortierella elongatula* | 0.08 | 0.20 | **0.75 \*\*** | −0.43 | 0.22 |
| *Ilyonectria liriodendri* | 0.12 | 0.33 | **0.80 \*\*** | −0.31 | 0.34 |
| *Lycogalopsis solmsii* | **0.63 \*\*** | −0.26 | −0.43 | 0.09 | −0.45 |
| *Ilyonectria destructans* | 0.16 | 0.37 | **0.80 \*\*** | −0.22 | 0.37 |

\* Shows the significance level at $p < 0.05$ and \*\* shows the significance level at $p < 0.01$. Bold also shows a significant correlation.

## 4. Discussion

Long-term consecutive monoculture leads to a severe decline in tea crop productivity, which has been already observed in previous studies [4,27,28]. Current research also confirms that a continuous tea cropping system significantly decreased the tea yield (Table 1). This phenomenon also occurs in both perennials and annual crops such as soybean, potato, apple, sugarcane, and black pepper, whose growth was severely hindered in a continuous monoculture system [6–10].

Soil physiochemical attributes such as pH are usually known as a fundamental factor for sustainable agricultural production. Our findings showed that continuous monoculture of tea significantly reduced soil pH (Table 2), which is consistent with previous findings [2,29]. Modern intensive farming systems based on the long-term application of inorganic fertilizers, especially nitrogen instead of organic fertilizers, can enhance the acidification of tea garden soils [30]. Another possible reason for soil acidification in tea consecutive monoculture systems is the different bioactive catechins produced by tea roots and leaves, which can act as allelochemicals, leading to soil acidification [4,31,32]. It has been reported that continuous monoculture of tea leads to accumulation of soil nutrient content (so-called nutrient sequestration) [33,34]. Consistently, consecutive monoculture of tea significantly increased the P content in the soil, which may result from the overuse of inorganic fertilizers [35,36], or from the fact that tea plants cannot use the available soil P efficiently in the long-term monoculture tea [37]. On the other hand, the reduction of AN, SOM and AP was observed in the T15Y field compared to fresh and old plantations (T2Y and T30Y) (Table 2). These results are consistent with previous findings in

which continuous tea growing systems result in the depletion of soil nutrients [2,33,34]. In summary, soil acidification, the accumulation of tea residues and allelochemicals, and the reduction of organic matter in a tea monoculture system can reduce tea yield.

Soil microbiome plays a significant role in soil function and ecosystem sustainability [4,38]. Investigating the response of soil fungal communities in consecutive cultivation systems can help us to understand the reasons for the reduced yield of long-term monoculture tea. Our study showed that species richness declines significantly as the tea planting years increase (Figure 2A–C), and these findings are consistent with previous studies. For instance, Zhao et al. [6] also revealed that soil fungal richness declined over the years of coffee planting duration. Similarly, Bai et al. [39] documented that fungal richness declined with increasing years of the consecutive planting of soybean under root-rot diseased soil. In this study, fungal diversity showed inconsistency; however, compared to CK, fresh and old cropped tea fields significantly decreased the fungal diversity (Figure 2D,E). These results are in line with previous findings, in which the fungal diversity decreased in response to continuous planting of *Panax notoginseng* [40]. The decrease in fungal diversity has been recognized as a foremost threat to ecosystem services [41], resulting in partial loss of soil function (plant growth promotion or disease inhibition) [42,43], which may result in reduced tea production under a consecutive monoculture system.

The results of UPGMA and PCoA in this study indicate that the continuous monoculture of tea strongly affects the soil fungal communities (Figure 3). Correspondingly, Xiong et al. [35] also observed a significant influence on alterations in the fungal community structure during continuous cultivation of vanilla. In fact, this phenomenon also occurs in both perennials and annual crops such as soybean [10,39], *Panax notoginseng* [40], potato [9], and coffee [6] continuous cropping systems. In addition, the UPGMA and PCoA helped to explain that the fungal communities fluctuated after 30 years of tea monoculture. Soil chemistry has an essential role in structuring microbial communities [38]. In combination with the RDA results (Figure 6);,we can hypothesize that a meaningful shift in the fungal community of T15Y and T30Y monoculture tea fields in terms of structure can be attributed to alterations in soil chemical features [38]. It is worthwhile to comment that the shift of fungal structure in response to monocultural tea systems could not be solely ascribed to differences in soil chemical properties, but may also be under long-lasting influences of tea plant root exudates or residues [4,44], which could not be determined within the scope of this study.

Soil-borne pathogens that cause various diseases in tea-growing areas around the world are considered to be significant problems, significantly reducing tea productivity and hindering the development of the tea industry [45,46]. The data obtained from Illumina sequencing presented us with some unique fungal taxa, such as plant beneficial or pathogenic fungi, which were suppressed or promoted in response to tea monoculture. Depending on their role in another ecosystem, we can also speculate on the ecological role of these fungal taxa in the continuous tea monoculture system [47]. In this light, our current attempt may help us to explain the consequences of consecutive monoculture on plant beneficial or pathogenic fungal taxa and consider their impacts on tea yield, which may provide essential information for future research. Our findings suggested that identified fungal taxa at the genus and species level, which are generally reported as plant pathogenic or beneficial, were suppressed or promoted in response to tea monoculture. It has been proposed that biocontrol microbes are a sustainable alternative to chemical control [48]. Some fungal genera, especially *Mortierella*, are fungal antagonists of plant pathogens, which help to suppress *Fusarium* wilt in banana plantations [49]. Furthermore, *Mortierella alpine* belonging to this genus suppresses corn rot disease, reduces biotic and abiotic stresses, and improves physiological parameters of saffron crocus plants [50]. In this study, the dominant genera such as *Mortierella* (*Mortierella alpine* and *Mortierella elongatula*) showed a decreasing trend in response to the monoculture of tea, which shows that continuous tea cropping decreases the possible density of beneficial fungi in the soils.

Several fungal species, such as *Fusarium oxysporum*, *Fusarium solani*, *Microidium phyllanthi*, *Ilyonectria liriodendri*, and *Dactylonectria pauciseptata* are potential pathogens that cause various

diseases in different crop plants. Among them, *Micoryium phyllanthi* originates powdery mildew on chamberbitter leaves [51], while *Ilyonectria liriodendri* and *Dactylonectria pauciseptata* also cause diseases, especially black foot and root rot in grapevine and plum, respectively. Similarly, *Gibberella intricans* is also known as a plant pathogen as it produces trichothecene in a wide range of plant species [52]. Likewise, *Fusarium oxysporum* and *Fusarium solani* cause root rot and wilt, collar canker, and dieback disease in tea plants [45,46]. Recent studies show that continuous planting of strawberry and soybean leads to an increase in the relative abundance of *Fusarium oxysporum* [53,54]. In this study, *Fusarium* (*Fusarium oxysporum*, *Fusarium solani*) and *Microidium phyllanthi* produced the same result as in a previous study in response to tea monoculture, suggesting that continuous tea cropping may increase the occurrence of pathogenic fungi in soils. However, *Gibberella intricans*, *Ilyonectria liriodendri*, and *Dactylonectria pauciseptata* showed a decreasing trend that indicates continuous tea cropping may suppress pathogenic fungi in soils. Suppression and accumulation of pathogenic fungi abundance at the genus and species levels were perceived in continuous tea fields, indicating that continuous tea planting has a dual impact on soil-borne pathogenic fungi. However, inhibition of plant-beneficial fungi suggests that continuous planting of tea reduces plant beneficial fungi in the soil. In this study, the decrease in tea yield in response to continuous cropping can be attributed to the reduction of beneficial fungi (*Mortierella alpine* and *Mortierella elongatula*) and the promotion of pathogenic fungi (*Fusarium oxysporum* and *Fusarium solani*).

## 5. Conclusions

Overall, our results demonstrated that low tea production under the long-term continuous cropping system may be associated with changes in fungal communities and soil physiochemical features, such as soil acidification, the decline in fungal species richness and potentially beneficial fungal communities, and the increase in potentially pathogenic fungi. This research grants us an invaluable avenue for progressing sustainable agricultural measures to enhance microbial activity and boost tea production in continuous cropping soils, which is essential for tea production in China.

**Supplementary Materials:** The following are available online at http://www.mdpi.com/2073-4395/9/8/466/s1, Figure S1: An operational classification unit (OTU) analysis of soil samples collected from the control and three-time series tea fields. The horizontal axis represents the sample name, the first vertical axis represents the tag number, and the second vertical axis represents the OTU number. "CK", "T2Y", "T15Y" and "T30Y" represent fields that have been continuously planted for 0, 2, 15 and 30 years, respectively, Figure S2: Box plots representthe relative abundance of fungal phyla in control and three series of tea fields. A: Zygomycota; B: Ascomycota; C: Basidiomycota; D: Glomeromycota; E: Chytridiomycota. Error bars with different lowercase letters indicate significant differences among treatments based on the LSD test ($p < 0.05$). "CK", "T2Y", "T15Y" and "T30Y" represent fields consecutively planted for 0, 2, 15 and 30 years, respectively, Figure S3: Box plots represent the relative abundance of top fungal generain control and three series of tea fields. Error bars with different lowercase letters indicate significant differences among treatments based on the LSD test ($p < 0.05$, * $p < 0.01$, *** $p < 0.001$)). "CK", "T2Y", "T15Y" and "T30Y" represent fields consecutively planted for 0, 2, 15 and 30 years, respectively, Figure S4: Box plots represent the relative abundance of top fungal species in the control and three series of tea fields. Error bars with different lowercase letters indicate significant differences among treatments based on the LSD test ($p < 0.05$, * $p < 0.01$, *** $p < 0.001$)). "CK", "T2Y", "T15Y" and "T30Y" represent fields consecutively planted for 0, 2, 15 and 30 years, respectively, Figure S5: Non-metric multidimensional scaling graphs (NMDS) showing differences of fungal communities between the analyzed soil samples obtained from control and three-time series tea plantations. "CK," "T2Y", "T15Y" and "T30Y" represent fields consecutively planted for 0, 2, 15 and 30 years, respectively. Table S1: Pearson's correlation between continuous cropping and abundant taxa of fungi (phyla, genera and species) in control and different age monoculture tea fields. * Shows the significance level at $p < 0.05$ and ** shows significance level at $p < 0.01$, Table S2: Species functions.

**Author Contributions:** W.L. and Y.A. conceived the study; Y.A., M.T., and W.L. wrote the paper; Y.A. and M.U.K. performed field sampling and lab experiments; Y.A., M.T., and H.A. performed the statistical analyses. All authors discussed the results and commented on the manuscript.

**Funding:** This research was supported by the National Key Research and Development Plan 2017YFE0121800, National Key Research and Development (R & D), China Plan 2016YFD0200900, and 948 programs from the Ministry of Agriculture (2014-Z360) China.

**Conflicts of Interest:** The authors declare no conflict of interest.

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
