# Peer review of "Long-Term Monoculture Negatively Regulates Fungal Community Composition and Abundance of Tea Orchards"

_agronomy, doi:10.3390/agronomy9080466_

Round 1
Reviewer 1 Report
The paper named “Long-term monoculture negatively regulate fungal community composition and abundance of tea orchards” present an interesting work on how monocropping cultivation of Tea orchards affect and rhizospheric soil physiochemistry and associated microbiota composition.
The work is overall well presented and it has been carried out with appropriate methodologies.
However, this article can be improved in different sections as following reported.
I also recommend a careful revision of the English language.

Reviewer 2 Report
Hello,
A nice little paper on fungal community responses to the long term monoculture of tea. Sadly thought the soil sampling strategy lacks clarity (e.g. how were random locations assigned, when were samples taken, how many samples were taken, what happened to the rhizosphere and bulk soil samples?). The methods are also often too scant to be reproducible. The results sometimes fall into discussion and the conclusions have pushed my buttons in that the presence of potentially pathogenic or beneficial fungi is sometimes stated as fact of their existence without Koch's postulates being undertaken. This needs to be played down and the presentation more mindful of this potential, rather than an actual. If you have disease incidence in the fields, then use that as a sign of pathogenic activity.
Additional comments as follows with some mark up on the manuscript.
Abstract
Not bad, but some of the word choice is probably not suitable. The link between pH and species is sold as driving the yield decrease, but then latter the statement is more appropriately measured (line 7 compared to 15). The clear differentiation of the fungi into good and bad concerns me without being appropriately balanced.
Introduction
The general flow is good, but there are still some issues with over simplification of facts. For example, not all Fusarium spp. are pathogenic, so is it fair to say this is the case? It’s certainly not correct. Some of the word choice also needs a bit of review.
50 - It’s a shame the aims of the paper are given and not the objectives, which would be more precise in definition. Aim ‘c’ is written as a hypothesis and not an aim.
Material and methods
57 - An average rainfall should not have a range of 1200 mm of rain. What is the average?
58 – so we have in field sampling and no planting replication. How are you accounting for adequate sampling and analysis? PCA has been mentioned so, whilst the distribution of the data, is not that important, an adequate approach to random population sampling is needed.
62 – how often are these fertilisers applied? If this is annual then there is your pH change, especially when linked to N offtake in the tea leaves.
65 – so even the non-cultivated field is fertilised?
68 – how was randomisation ensured? It is not as easy as just dropping in your fork when you want to. When were samples taken? How many samples?
75 – utilising hydrochloric acid and ammonium fluoride is not enough detail for the extraction and analysis, especially without a reference.
77 - reference for the N method or more detail.
Were all the fungal sequences assigned a classification? If not then at what point, if ever, where unknowns removed?
Results
115 - The first results is for yield. No method as to how or when yield was measured. I personally hate this way of cross referencing to a figure or table as it is contrary to what I was taught. The last sentence is also discursive, so is this a results and discussion section?
Figure 1 will need to be of a higher quality.
119 – mean of how many samples? Is this over a number of years? LSD, so an unbalanced ANOVA if we have 2, 15 and 30 years of yield data? If not is this of a series of sub samples within each field in a given year?
Table 1 – how many samples and from when? The legend appears to be under the table, as a footnote. I’d move this to the legend.
Be consistent with your numbering. No commas and space between hundres and thousands or no space, but not a bit of everything.
How many sequencing samples and from when? We’ve bulk and rhizosphere soil as well. Were these evenly represented?
Is your Simpsons actually Simpson or Gini-Simpson? I am surprised that it follows the Shannon index if it is truly Simpson.
142 – discursive again, although an interesting point. Why is the diversity (or likelihood of similarity), so much lower at 15 years over 30? Given the yield data (figure 1) was similar for 15 and 30 years it would have me start to query some of your statements on effect.
Lovely clustering in figure 4. Looks like there were four samples in each field site from this, not that this is conformed anywhere else in the manuscript. Ordination with less than five factorial representatives often provides warnings on the confidence of the results. Whilst these look good it is worth noting that one more rep (even if a within field pseudoreplicant) would have added more confidence to the ordination groupings and may have allowed PERMANOVA to be undertaken to confirm statistical divergence. Is that two rhizosphere and two bulk soil samples?
182 – why not start the genera summation with a new paragraph having covered phyla?
178 – ten most prominent genera, but figure 5 only has eight?
Line numbers have done something horrible in Table 2. Interesting that the only moderate relationship is between the Mortierella and pH. The rest are all weak to none, even when significant.
Discussion
264 – pH and P. Come on, the link there is far more likely if you have a low Fe and Al soil. Of course there are not sufficient details in this paper to determine that, but it is more likely that the pH shift, for whatever reason (and those given are fine) is responsible to the increase in available P.
263 and 269 are at odds. Does tea promote nutrient accumulation or nutrient mining?
279 – it is this inconsistency that I find interesting. Are you going to explore or offer a reason for it?
295 – Sorry, figure 7 clearly shows that your soil chemical differences do support fungal community shifts. Given period of monoculture between replanting could be assigned a numerical value, it could be included in a PCA with soil factors. Have suggested what you may have meant.
Line 301 – as someone who has worked as and with pathologist I struggle with this flippant categorising of pathogens and beneficials. Without Koch’s postulates it has to be presented as a potential benefical or pathogen. That the rules. Line 302 is therefore appropriate.
333 – this is a highly speculative statement given there is no direct evidence that the fungi you saw an increased abundance of (bearing in mind this is sequence and not biomass data) are actually beneficial to tea production in these soils.
335 – Can it really? Have you read any of Rishiraj Dutta’s work? He’s never looked at the fungi, but his correlations with rain, N fertiliser and age of the plantation are far stronger than yours. His sampling strategies are also far more transparent and generally multi seasonal. Have a look:
https://www.researchgate.net/publication/250104416_Effects_of_Plant_Age_and_Environmental_and_Management_Factors_on_Tea_Yield_in_Northeast_India
https://www.researchgate.net/publication/231755165_Analysis_of_factors_that_determine_tea_productivity_in_Northeastern_India_A_combined_statistical_and_modelling_approach
I can’t see anywhere in this paper where this statement is clearly supported. Perhaps an ordination using a grouped potential beneficial and pathogen clustering against yield is needed.
340 – says ‘might’, which just goes to back up my previous point.

Reviewer 3 Report
Summary
The authors tried to understand why tea yield decreases during tea planting periods by monitoring fungal communities by ITS meta sequences.
They have surveyed three different fields, which keeps tea planting for 2, 15, and 30 years, respectively. Although the soil nutrient conditions among them does not seem to significantly change, they found that species riches of fungal communities in 15 years and 30 years differ from those in 2years. Actually species riches decrease in the longer planting period (although the graph image in the corresponding figure is not clear due to the low resolution…).
Although some of the images are not clear (the authors should provide better images!), the works still provide valuable information to think about the effect of continuous monocultural agriculture in the context of microbiome.
Minor points
・It is normally difficult to conclude just from ITS information that the particular fungus detected is whether pathogenic or beneficial in the condition (e.g., Hiruma et al., current opinion 2018) . It would be nice to carefully discuss the issues in this manuscript.
・Please prepare the better images.
・It would be better if the authors would provide a bit more regends in each figure.
Round 2
Reviewer 2 Report
Thank you for the quick turnaround.
The abstract reads much better and you have played down the overstatement of pathogens much better. There is still a part of me that says simply associating a fungal species to a plant disease is inappropriate, but perhaps 25 years of plant pathology has done that to me.
An average rainfall is still not a range and that is a big range. I've suggested changes.
The sampling is a little clearer, although how the random plants were selected is still a mystery. Shame you pooled five into one and not four, would have given your ordinations more power. How is the CK sample comparable if there are no trees to sample? We often struggle with this one when comparing replanted with pasture and I am sure others do.
There is now something on the tea yield analysis, but it is still thin. Was the entire crop picked, just the sampled plants? There is more to this that is needed if the work was to be replicated.
The soil analysis is better now I know which resource it is based off.
Results. Given the g is the weight from which the kg/ha is derived you don't need both. Pick one and report one. Please also superscript all the significant differences in tables, it makes reading the numbers easier. I would also question whether two decimal points are valid and I still don't like the A in the N, P and K determinations. Being a bit fussy so must be having a bad day. Sorry.
The figures have all come across poorly, but the supplemental ones look great. May be an editorial issue. I am not convinced the reader needs to see a rarefaction curve data and if the table in S2 is so important, then why not have it in the paper?
Discussion. There is something about the T15Y that just does not quite add up. Why is it so similar to the adjacent field (CK)? Is that strip of tea on a slightly different soil type or a change in the topography? This is why sampling from continuous plots is challenging even if you argue out of pseudoreplication and have truly randomised samples.
This treatment also features as an anomaly in the fungal DNA analysis. Sure there is a decline from CK to T2Y to T30Y, but why does T15Y appear to be less fungally diverse and community stable than T30Y? I think you need to look at your potentially beneficial and pathogenic fungi closer. Determine if there is a shift one way or another between T2, T15 and T30 and perhaps see if it is plausible that potential pathogens may develop or a loss of beneficials occur under T15, and then if there is not a reversal in this and therefore a potentially supressive system development under T30. Sure this is all hypothetical without disease records (or rather symptoms), but this is the obvious nugget of real genius that may come out of this work. Why is T15 different? Without it this is just a further extension of what has already been reported.
